# Neutrophil Extracellular Traps in Skin Diseases

**DOI:** 10.3390/biomedicines9121888

**Published:** 2021-12-12

**Authors:** Youichi Ogawa, Yoshinori Muto, Manao Kinoshita, Shinji Shimada, Tatsuyoshi Kawamura

**Affiliations:** Department of Dermatology, Faculty of Medicine, University of Yamanashi, Koufu 409-3898, Yamanashi, Japan; ymuto@yamanashi.ac.jp (Y.M.); mkinoshita@yamanashi.ac.jp (M.K.); sshimada@yamanashi.ac.jp (S.S.); tkawa@yamanashi.ac.jp (T.K.)

**Keywords:** neutrophils, neutrophil extracellular traps, psoriasis, generalized pustular psoriasis, acute generalized exanthematous pustulosis, Behcet’s disease, pyoderma gangrenosum, hidradenitis suppurativa, Stevens-Johnson syndrome/toxic epidermal necrolysis

## Abstract

Neutrophils are the primary innate immune cells, and serve as sentinels for invading pathogens. To this end, neutrophils exert their effector functions via phagocytosis, degranulation, reactive oxygen species generation, and neutrophil extracellular trap (NET) release. Pathogens and pathogen-derived components trigger NET formation, leading to the clearance of pathogens. However, NET formation is also induced by non-related pathogen proteins, such as cytokines and immune complexes. In this regard, NET formation can be induced under both non-sterile and sterile conditions. NETs are enriched by components with potent cytotoxic and inflammatory properties, thereby occasionally damaging tissues and cells and dysregulating immune homeostasis. Research has uncovered the involvement of NETs in the pathogenesis of several connective tissue diseases, such as systemic lupus erythematosus, rheumatoid arthritis, and ANCA-associated vasculitis. In dermatology, several skin diseases clinically develop local or systemic sterile pustules and abscesses. The involvement of neutrophils and subsequent NET formation has recently been elucidated in these skin diseases. Therefore, this review highlights the NETs in these neutrophil-associated diseases.

## 1. Introduction

Neutrophils are the first line of defense against invading pathogens. They exert effector functions via phagocytosis, degranulation, reactive oxygen species (ROS) generation, and neutrophil extracellular trap (NET) release. NET release is triggered by various physiological stimuli, such as pathogens, pathogen-associated molecular pattern molecules, cytokines, immune complexes, cholesterol, and microcrystals in vivo. Thus, pathogen clearance by neutrophil-derived NET is vital for living beings, including humans and even plants. On the other hand, neutrophils could release NETs even in the absence of pathogens. Given that NET components contain numerous potent immune modulators such as histones, granule-derived proteases, and antimicrobial peptides, NET release under sterile conditions triggers immune dysregulation and causes tissue damage [1,2]. NETs were reportedly associated with the pathophysiology of several connective tissue diseases such as systemic lupus erythematosus, rheumatoid arthritis, and ANCA-associated vasculitis [3]. In the dermatological field, NET research has been conducted in psoriasis [4,5]. However, there are various neutrophil-associated skin diseases, including neutrophilic dermatosis and pustular dermatosis. Therefore, this review sought to update the involvement of NET in the pathomechanisms of dermatological neutrophil-associated diseases.

## 2. Neutrophil Extracellular Traps

NETs were first reported in the literature by Takei et al. in 1996 [6], and their details and functions were subsequently investigated by Arturo Zychlinsky’s laboratory [7]. NET release can be defined as the release of modified chromatin decorated with granular proteins, nucleus, and cytoplasm [1,2]. Given that NET release was initially identified as accompanying neutrophil death, the process of NET release was termed NETosis [8]. However, NET formation has been clarified as being triggered by three different causes in accordance with NADPH dependency and neutrophil life cycle: NADPH oxidase-dependent NETosis; NADPH oxidase-independent, mitochondrial ROS-dependent NETosis; and non-lytic chromatin release. In this regard, perhaps the third type of NET formation should not be called NETosis [2]. As described here, NETosis was first termed based on the origin of cells (neutrophils). However, eosinophils, mast cells, monocytes, and macrophages can also release extracellular traps (ETs). Thus, cell death with release of ETs was renamed as ETosis [1,2].

Considering that the first type of NET formation has been examined in detail, the underlying mechanism of NADPH oxidase-dependent NETosis induced by phorbol 12-myristate 13-acetate (PMA) is described. PMA activates protein kinase C, followed by phosphorylation of NADPH oxidase subunits [9], leading to the generation of ROS such as superoxide anion radical and hydrogen peroxide (H_2_O_2_). Therefore, NETs will not be released when there is pharmacological inhibition of NADPH oxidase or patients have mutations in NADPH oxidase [10,11]. There are eight types of proteins in the azurosome: myeloperoxidase (MPO), neutrophil elastase (NE), cathepsin G (CatG), azurocidin, lactoferrin, proteinase 3 (PR-3), lysozyme, and eosinophilic cationic protein. Among these proteins, NE, CatG, and azurocidin are homologous serine proteases [12]. H_2_O_2_ dissociates azurosome, followed by a release of serine proteases as well as MPO into the cytosol and subsequent migration into the nucleus [12]. In the nucleus, histones are citrullinated by peptidyl arginine deaminase 4 (PAD4), leading to a weakened connection of histones with chromatin, and NE dissociates histones from chromatin in cooperation with MPO, resulting in chromatin decondensation [13]. Finally, nuclear, granular, and cytoplasmic membranes are disrupted by a pore-forming protein, gasdermin D [14], followed by the release of chromatin together with various antimicrobial molecules, including histones, NE, MPO, and antimicrobial peptides into the extracellular space [15]. Of note is that the mechanism described above is one example mediated by PMA. We have to consider that there are three means in which NET is formed, and NET formation is dependent on the type of stimuli.

In humans, at least two types of circulating neutrophils have been reported: conventional polymorphonuclear neutrophils (PMNs) and low-density granulocytes (LDGs). In contrast to PMNs, LDGs exhibit higher staining for NE and lower staining for its inhibitor secretory leukocyte proteinase inhibitor (SLPI) [16]. However, this may be one of the reasons that LDGs are prone to NETosis, thereby contributing to the pathomechanisms of various autoimmune and autoinflammatory diseases. LDGs are a heterogenous population consisting of both immature and mature neutrophils [17]. Clinically, patients with autoimmune diseases, such as systemic lupus erythematosus (SLE) and rheumatoid arthritis (RA), exhibit an increased number of circulating LDGs. Moreover, these LDGs not only undergo NETosis but also exhibit an enhanced ability to synthesize tumor necrosis factor-α (TNF-α) and interferons (IFNs), thereby facilitating tissue inflammation and damage [18,19]. In an experimental setting, PMNs settle together with red blood cells during density gradient preparation. In contrast, LDGs remain at the peripheral blood mononuclear cell (PBMC) layer due to their buoyancy [20].

## 3. NETs in Skin Diseases

There is increasing evidence regarding NET involvement in the pathomechanism of several skin diseases [21]. Although there are various skin manifestations in patients with SLE, RA, and ANCA-associated vasculitis, descriptions about these connective tissue diseases are omitted, because there are already excellent reviews regarding the association between these diseases and NETs [1,21].

### 3.1. Psoriasis

Psoriasis is a chronic systemic inflammatory disease (Figure 1). It primarily affects the skin, but can also affect joints and internal organs. Psoriasis is mediated by the interleukin (IL)-23/T-helper (Th)17 axis. In this axis, IL-23 is mainly produced by dermal dendritic cells (DCs). Subsequently, IL-23 differentiates Th17 cells in coordination with other cytokines. Pathologically, the predominant dermal infiltrates in psoriatic skin lesions are lymphocytes, not neutrophils. However, the presence of subcorneal Munro’s microabscesses filled with neutrophils is one of the pathological hallmarks of psoriasis. In addition, neutrophils might not be the central player in the pathomechanism of the disease. However, research regarding psoriasis and neutrophils have revealed the close association between psoriasis and neutrophils [4].

The circulating neutrophils of patients with psoriasis are more prone to NETosis, either spontaneously or in response to lipopolysaccharides, compared with those of healthy volunteers [22,23,24,25]. Moreover, the amount of NETotic cells in the peripheral blood correlates with the severity of psoriasis [24]. Consistent with these data, the number of circulating LDGs is increased in patients with psoriasis, similar to patients with SLE, and correlates with the severity of psoriasis [26]. The sera from psoriasis patients induce NET formation in healthy neutrophils [24].

Exosomes derived from human epidermal keratinocytes treated with psoriasis-related cytokines, such as IL-17A, IL-22, IFN-γ, and TNF-α, stimulate normal human neutrophils, followed by NF-κB and p38 MAPK signaling activation, leading to the production of TNF-α, IL-6, and IL-8, and NET formation [27]. In line with this in vitro experiment, NETotic neutrophils are present in both the epidermis [24,25] and the dermis [25].

In two human ex vivo psoriasis-like skin models, topical leukotriene B4 application and tape-stripping that share some histological features with psoriasis, neutrophils and T cells infiltrate the skin. Surprisingly, staining for IL-17 protein and IL-17 mRNA reveals that the majority of the IL-17-expressing cells are neutrophils and mast cells (MCs). Moreover, T cells represent a minority of the IL-17-expressing cells. Neutrophils, but not MCs, co-express the IL-17-associated transcription factor RORγt and form NETs. The number of MCs during the inflammatory process is steady, whereas the number of neutrophils is dynamic over time [28]. Consistent with these data, epidermal NETotic neutrophils in Munro’s microabscess co-localize with IL-17A. Additionally, some of the dermal NETotic neutrophils and MCs that undergo MCET formation (MCETosis) also co-localize with IL-17A and LL-37, suggesting that ETotic neutrophils and MCs may serve as sources of IL-17A and LL-37 [25]. In turn, IL-17A produced by ETotic neutrophils and MCs facilitates further neutrophil accumulation [29]. As reference, in vitro culture of normal human skin explants with IL-1β and IL-23 induces MC degranulation and MCETosis [25].

In vitro culture of anti-CD3/CD28 bead-treated PBMCs with NETotic neutrophils promotes Th17-cell differentiation [30,31]. Given that CD15^+^, CD10^−^, and CD66b^low^ neutrophils accumulate close to T cells in the upper dermis of psoriatic skin [32], NETs-mediated Th17-cell induction might be occurring in vivo. Additionally, in vitro cultures of human epidermal keratinocytes with NETotic neutrophils from patients with psoriasis induces human β defensin-2 mRNA and protein expression [24]. These in vitro experiments suggest that NETotic neutrophils in psoriasis are involved in the establishment of immunological features of psoriasis.

Psoriatic skin contains complexes of NET-associated deoxyribonucleic acid (DNA), CatG, and SLPI. DNA/CatG/SLPI complexes trigger the production of type I IFNs by human plasmacytoid dendritic cells (pDCs) via endosome-localized receptors. In this process, CatG is central in activating the pDCs by allowing toll-like receptor (TLR) 9 to sense extracellular self-DNA. However, to achieve CatG-mediated DNA delivery to TLR9, SLPI is required, because in the absence of SLPI, the synthesis of type I IFNs by pDCs is greatly impaired [33,34]. Additionally, psoriatic skin also contains complexes of NET-associated ribonucleic acid (RNA) and LL-37. RNA alone does not induce IL-8 production by neutrophils in vitro. However, LL-37 facilitates RNA uptake by PMNs and directs RNA to cytoplasmic compartments where nucleic acid-recognizing intracellular TLRs are located, leading to the production of proinflammatory cytokines such as TNF-α, IL-6, IL-1β, and macrophage inflammatory protein (MIP)-1β. Moreover, the RNA and LL-37 complex induces NETs in normal neutrophils, thereby amplifying NET-associated inflammatory responses [35]. Collectively, complexes of NET-associated nucleic acid and components subsequently activate neutrophils and pDCs.

A fumaric acid-enriched plant, *Fumaria officinalis*, has been empirically used for the treatment of inflammatory skin diseases since the 17th century. The clinical use of fumaric acid derivatives started in the late 1950s [36]. Several decades later, a more standardized derivative was approved for the treatment of psoriasis in Germany [37]. Interestingly, dimethylfumarate inhibits neutrophil activation, including ROS production, NET formation, and migration in vitro [38], suggesting that the effect of fumaric acid in inflammatory skin diseases in vivo is mediated through an inhibition of neutrophil activation.

Taken together, neutrophils can contribute to the pathomechanism of psoriasis as a source of cytokines and chemokines, including IL-17, through NET formation and subsequent differentiation of Th17 cells and activation of pDCs and adjacent neutrophils (Figure 2).

### 3.2. Pustular Dermatosis

Pustular dermatoses are defined as skin diseases in which sterile pustules are clinically visible on the skin, in contrast to psoriasis. The size of pustules varies between the diseases, and they are pathologically filled with neutrophils. Several skin diseases are categorized into pustular dermatoses, such as generalized pustular psoriasis (GPP), impetigo herpetiformis (IH), acral pustular psoriasis (APP), acute generalized exanthematous pustulosis (AGEP), acute generalized pustular bacterid, or subcorneal pustular dermatosis (SPD) (Figure 1). Among these pustular dermatoses, GPP, IH, and APP are postulated as psoriasis-related diseases. GPP and IH patients develop high fever, general malaise, and generalized erythroderma along with disseminated sterile pustules. GPP could develop in patients with or without a history of psoriasis. The classic type of IH is a variant of GPP that develops during the third trimester of pregnancy and spontaneously resolves after delivery. APP is an acral type of psoriasis with relatively large sterile pustules, and often develops extra-palmoplantar lesions resembling psoriasis. AGEP is a severe cutaneous adverse drug reaction that has similar symptoms with GPP or IH; it is impossible to discriminate GPP, IH, and AGEP based on clinical symptoms alone.

IL-36s (α, β, and γ) and IL-36 receptor antagonists (IL-36Ra) belong to the IL-1 cytokine family and compensate each other to maintain inflammatory homeostasis. Loss-of-function mutations in *IL36RN*, which encodes IL-36Ra, cause a recessively inherited autoinflammatory keratinization disease known as deficiency of IL-36Ra (DITRA) (IL36RN [MIM: 605507]) [39,40,41,42], because IL-36Ra’s role is to suppress excessive IL-36 signaling. Consistently, IL-36Ra-deficient mice treated with imiquimod exhibited severe epidermal proliferation and dermal neutrophilia along with NET formation [43,44,45]. Importantly, neutrophils are engaged to activate IL-36 signaling. Inactive pro-IL-36s secreted from keratinocytes and dermal DCs are required to be proteolytically processed by neutrophil granule-derived proteases, including CatG, elastase, and PR-3, for their activation [46,47]. In this regard, NETotic neutrophils efficiently execute this processing [47,48]. DITRA reportedly includes GPP “without” a history of psoriasis, AGEP, and IH [41,49,50,51,52].

*MPO* encodes the heme-containing enzyme MPO that is the major protein in the azurosome of azurophilic granules. MPO is primarily found in neutrophils and, to a lesser extent, in monocytes. In the presence of H_2_O_2_, which derived from NADPH oxidase during the respiratory burst, MPO catalyzes the generation of strong reactive intermediates, including hypochlorous (HOCl), hypobromous and hypothiocyanous acids, tyrosyl radicals, and reactive nitrogen intermediates [53,54]. In particular, MPO/HOCl is critical for the intracellular killing of some bacteria and fungi by neutrophils [54]. Thus, MPO deficiency (MPOD [MIM: 254600]) represents an immune deficiency due to defective intracellular killing of pathogens. *MPO* mutations are also identified in patients with pustular dermatoses, such as GPP, APP, and AGEP [55,56]. MPO-deficient neutrophils exhibit an enhanced enzymatic activity of serine proteases, including CatG, elastase, and PR-3, that activate IL-36s, leading to the enhanced inflammatory condition. NET formation is impaired in MPO-deficient neutrophils compared with those of normal neutrophils. Additionally, MPO-deficient neutrophils exhibit higher CD47 expression, which is a “don’t eat me” signal, resulting in impaired monocyte–mediated phagocytosis of neutrophils (efferocytosis). Together, MPO mutations trigger excessive IL-36-mediated inflammation and allow neutrophils to stay in situ [55].

When focused on GPP, 19–41% of GPP cases develop as DITRA [57,58]. However, several variants of *MPO* mutations are identified in GPP patients without IL-36RN deficiency. Of note is that mutations in both *IL-36RN* and *MPO* result in enhanced IL-36s activity. In this context, IL-36 might be central in both the local formation of visible sterile pustules and severe systemic inflammation, such as high fever, general malaise, and generalized erythroderma, observed in patients with GPP and AGEP.

### 3.3. Behcet’s Disease

Behcet’s disease (BD) is a systemic vasculitis affecting multiple organs, including the skin, mucosa (recurrent oral and genital ulcers), eyes, joints, intestines, arteries, and central nervous system (Figure 1). The pathogenesis of BD is neutrophil hyperactivation. In addition to dysregulation of innate immunity, activation of Th1, Th17, and Th22 as well as impaired regulatory T-cell function and IL-10 expression has been reported, suggestive of dysregulation of acquired immunity [59].

Circulating neutrophils from patients with BD are prone to spontaneous NETosis in vitro [60,61,62,63]. Moreover, neutrophils from patients with active BD exhibit higher PAD4 levels and NE and ROS release compared to those from healthy subjects and inactive BD [60,64].

Sera from patients with active BD with vascular involvement contain elevated levels of cell-free (cf)DNA and MPO-DNA complexes compared with those without vascular involvement, inactive BD, and healthy subjects. Levels of cfDNA and MPO-DNA complexes are correlated with thrombin generation in the plasma of BD patients. Interestingly, deoxyribonuclease (DNase) treatment suppresses thrombin generation in the plasma of BD patients but not in that of healthy subjects [63]. Exposure to sera from patients with BD to the circulating neutrophils from healthy subjects induces oxidative burst and NADPH oxidase protein expression, followed by NET formation and enhanced *PAD4* mRNA expression, suggesting that sera from patients with BD contain certain soluble factors to facilitate NETs in normal neutrophils [60,61]. One candidate is soluble CD40 ligand (sCD40L), because sCD40L is enriched in the active BD sera compared with inactive BD sera, and sCD40L blockade of active BD sera attenuates NET formation [61]. Additionally, endothelial cells cultured with NETotic neutrophils from patients with BD decrease their proliferation and promote their apoptosis and cell death, implying NET involvement in BD vasculitis [60]. Macrophages cultured with NETotic neutrophils from patients with BD induce production of IL-8 and TNF-α and differentiation of IFN-γ-producing CD4^+^ T cells [62].

Indeed, NETotic neutrophils infiltrate blood vessel walls of the dermis and subcutaneous tissue [60,63,65]. Moreover, immunohistological studies have shown that IL-17A-producing CD4^+^ and CD8^+^ T cells, possibly primed by CD11c^+^ DCs and CD68^+^ macrophages, also accumulate perivascularly. In addition, IL-17A co-localizes with NET structures. There are two possibilities for explanations. One is that NETotic neutrophils release IL-17A, as previously reported. The other is that T-cell-derived IL-17A recruits and activates neutrophils followed by NET formation [65].

Collectively, NETs appear to be involved in the pathogenesis of vasculitis observed in patients with BD. Thus, targeting NETs may be a promising therapeutic strategy for the amelioration of BD-associated vasculitis and thrombosis. Indeed, NET formation in BD neutrophils is suppressed by colchicine and dexamethasone, which are clinically used for the treatment of BD as well as by Cl-amidine (a specific PAD4 inhibitor) and NAC (a ROS inhibitor) [60,64]. Apremilast, a small molecule inhibitor of phosphodiesterase 4, is the only specifically licensed drug for the treatment of BD. Hence, the effect of apremilast on NET formation should be addressed.

### 3.4. Pyoderma Gangrenosum

Pyoderma gangrenosum (PG) is a type of cutaneous neutrophilic dermatoses characterized by massive neutrophil infiltration in the affected skin devoid of infection and vasculitis (Figure 1) [66]. A remarkable efficacy of granulocyte and monocyte adsorption apheresis for the treatment of PG reinforces the involvement of neutrophils in the pathogenesis of PG [67]. Approximately half of PG cases are associated with systemic disorders, such as inflammatory bowel disease, monoclonal gammopathy, hematologic malignancy or paraproteinemia, BD, Sweet’s syndrome, hepatitis, human immunodeficiency virus infection, SLE, pregnancy, and Takayasu arteritis [66]. It is interesting to note that NETs involvement has been reported in some PG-associated underlying disorders. Circulating neutrophils from patients with PG spontaneously undergo NETosis. Moreover, over half of neutrophils in PG lesions exhibit NET formation [68,69,70]. Partial colocalization of IL-1β and TNF-α in NETs implies their contribution into NET formation [69]. Additionally, *MPO* mutations have been reported in PG [56,71].

PG is one of the major conditions that consist of pyogenic sterile arthritis, PG, and acne (PAPA) syndrome (PAPAS [MIM: 604416]). PAPA syndrome is a rare autosomal dominant disorder caused by gene mutations of *proline/serine/threonine phosphatase-interaction protein 1* (*PSTPIP1/CD2BP1*), resulting in the aberrant activation of innate immune systems followed by excessive overproduction of proinflammatory cytokines such as IL-1β, TNF-α, IL-6, IL-17A, and IFN-γ [72]. Therefore, PAPA syndrome is referred to as an autoinflammatory disorder. PBMCs from patients with PAPA syndrome contain more LDGs than those of healthy subjects. Moreover, these LDGs are prone to NETosis. Sera from patients with PAPA syndrome induce NET formation in neutrophils from healthy subjects in an IL-1-dependent manner. Additionally, neutrophils from patients with PAPA syndrome undergo NETosis in the presence of recombinant IL-1β, whereas those from healthy subjects do not, suggesting that PAPA neutrophils are primed to respond to IL-1β. As expected, NETotic neutrophils are present in the active PG lesions of patients with PAPA syndrome [73].

### 3.5. Hidradenitis Suppurativa

Hidradenitis suppurativa (HS), also refereed as acne inversa, is a chronic cutaneous inflammatory disorder characterized by persistent and recurrent abscess-like subcutaneous nodules and sinus tracts with purulent discharge that affects the axillae, buttocks, groin, and anogenital region (Figure 1). Oral tetracyclines are often administrated to control the bacterial load and for their anti-inflammatory effects, but their efficacy is not sufficient in patients with severe HS. The association and involvement of IL-17A produced by Th17 cells, which facilitates neutrophil migration and subsequent tissue damage, have been highlighted, [74,75].

Circulating neutrophils from patients with HS spontaneously undergo NETosis. However, NET complexes are not elevated in the sera of patients with HS. Moreover, the sera of patients with HS are unable to degrade NETs induced in healthy neutrophils. NET formation is enhanced in circulating neutrophils of HS patients, and some defective mechanisms for degrading NETs underlie this phenomenon. NETotic neutrophils are present in HS lesions, particularly in the lesional tunnel, and the degree of NETs and the severity of HS are positively correlated. Interestingly, autoantibodies against citrullinated proteins derived from NET components have been detected in patients with HS [76,77].

### 3.6. Other Neutrophil-Related Skin Disorders

Lesional skin neutrophils form NETs in approximately 40% of patients with Sweet’s syndrome and SPD [69,70]. Partial colocalization of IL-1β and TNF-α in NETs of Sweet’s syndrome implies their contribution into NET formation [69]. MPO deficiency has also been reported in patients with Sweet’s syndrome [71].

### 3.7. Stevens-Johnson Syndrome and Toxic Epidermal Necrolysis

Stevens-Johnson syndrome (SJS) and toxic epidermal necrolysis (TEN) are life-threatening mucocutaneous adverse drug reactions (cADRs) characterized by massive epidermal detachment (Figure 1). Cytotoxic CD8^+^ T cells and associated effector molecules, such as soluble FasL [78,79], perforin/granzyme B [80], granulysin [81], and IL-15 [82], are known to drive SJS/TEN pathophysiology. On the other hand, the contribution of innate immunity in the disease pathophysiology of SJS/TEN has been largely unexplored. However, the presence of NETotic neutrophils in both the epidermis and dermis of lesional SJS/TEN skin has been elucidated [82]. Lipocalin-2 (LCN-2)-derived from drug-specific CD8^+^ T cells triggers NET formation in neutrophils in the infiltrated skin. Subsequently, NETotic neutrophils also release LCN-2, which amplifies NET formation in a paracrine fashion. Moreover, LL-37-derived from NETs induces the expression of formyl peptide receptor 1 (FPR1), a family of G protein–coupled receptors, on SJS/TEN keratinocyte surfaces [83]. Keratinocyte death in SJS/TEN involves necroptosis, a form of programmed cell death. Necroptosis is mediated by the binding of monocyte-derived annexin A1 to FPR1, which is expressed on SJS/TEN keratinocytes [84]. Lastly, LL-37 is also released from keratinocytes undergoing necroptosis, followed by further FPR1 induction on adjacent keratinocytes, potentially enhancing the LL-37–FPR1–annexin A1 axis during SJS/ TEN disease progression [83,84].

Circulating neutrophils from patients with SJS/TEN are prone to spontaneous NETosis, but not those from patients with non-severe cADRs. The degree of spontaneous NETosis in the circulating SJS/TEN neutrophils is comparable with or greater than the circulating septic neutrophils [83].

Sera from patients with SJS/TEN, but not sera from patients with other types of cADRs, induce NET formation in normal neutrophils. Interestingly, blister fluids from SJS/TEN lesions also induce NET formation in normal neutrophils. Serum levels of NET-associated dsDNA, LL-37, and MPO-DNA complex are exclusively elevated in patients with SJS/TEN, but not in patients with other types of cADRs. These data suggest that NET formation is a specific phenomenon that occurs in SJS/TEN among cADRs [83].

More importantly, sera from patients with SJS/TEN induce much stronger NET formation than those from patients with SLE, psoriasis, GPP, and PG. Moreover, serum levels of NET-associated dsDNA, LL-37, and MPO-DNA complex are also much higher in sera from patients with SJS/TEN than those from patients with SLE, psoriasis, GPP, and PG [83].

Collectively, these data propose an additional mechanism that underlies SJS/TEN onset and progression, wherein the causative drugs trigger the orchestration of CD8^+^ T cell, neutrophil, and monocyte–mediated keratinocyte necroptosis through a pathway that is centered on NETosis.

## 4. Conclusions

Neutrophil-associated dermatological diseases are clinically categorized into four groups: erythema without visible pustules (psoriasis and SJS/TEN); erythema with visible pustules (GPP, IH, APP, AGEP, and SPD); neutrophilic dermatoses with ulcers and scars (PG and HS); and tender and indurated erythema (BD and Sweet’s syndrome) (Figure 1). The degree and layer of neutrophil infiltration varies between groups. Of note is that the presence of NETotic neutrophils is not evident in the pustular psoriasis, such as GPP and AGEP, but aberrant cutaneous neutrophilia is clearly observed in these diseases. Given that *MPO* mutations leading to impaired neutrophil clearance have been identified in some pustular dermatites such as GPP, APP, and AGEP, neutrophils in these diseases may be just standing in situ as a consequence of dysregulated inflammation and subsequent neutrophil recruitment. On the other hand, the presence of NETotic neutrophils is evident in other disease types. In psoriasis, NETotic neutrophils amplify the inflammation spirals. In SJS/TEN, they promote keratinocyte necroptosis. In PG and HS, they can damage dermis and subcutaneous tissue, resulting in the formation of ulcers and/or scars. In BD and Sweet’s syndrome, they also damage blood vessels or the dermis, resulting in the formation of tender and indurated erythema. However, the underlying immunological differences leading to the different clinical features are still unknown. Nowadays, three different processes of NET formation have been proposed in accordance with NADPH dependency and the neutrophil’s life cycle. The involvement of NET formation in each disease should be examined in future studies, which may uncover and make available new aspects of NET research. Skin manifestations are obvious, and skin samples are readily available. In this regard, neutrophilic dermatosis is a suitable area when researching neutrophils and NETs.

NET formation is likely to be involved in the pathophysiology of neutrophilic dermatoses. Hence, the inhibition of NETs is a promising therapeutic option. One of the biggest candidate medications is recombinant human DNaseI, which has been known to degrade NETs. It is noteworthy that administration of recombinant human DNaseI is clinically approved for the treatment of cystic fibrosis [85]. NET degradation is one of the underlying mechanisms for the efficacy of DNaseI in cystic fibrosis [86]. The efficacy of recombinant human DNaseI in reducing NETs has also been recently recognized in the treatment of chronic thromboembolic pulmonary hypertension [87]. An alternative approach for the treatment of NET-associated diseases is to inhibit the upstream molecules that trigger NET formation. For example, exosomes from in vitro-stimulated keratinocytes with psoriasis-related cytokines induce NET formation in normal neutrophils. In BD, sCD40L is a candidate for NETs induction. In SJS/TEN, LCN-2, produced by drug-specific CD8^+^ T cells, induces NETs in normal neutrophils, and in turn NET-associated LCN-2 and LL-37 further induce NETs in a paracrine fashion. Although research in terms of NET triggers is not sufficient, the development of agents to inhibit the above-mentioned molecules will improve treatment efficacy.

## Figures and Tables

**Figure 1 biomedicines-09-01888-f001:**
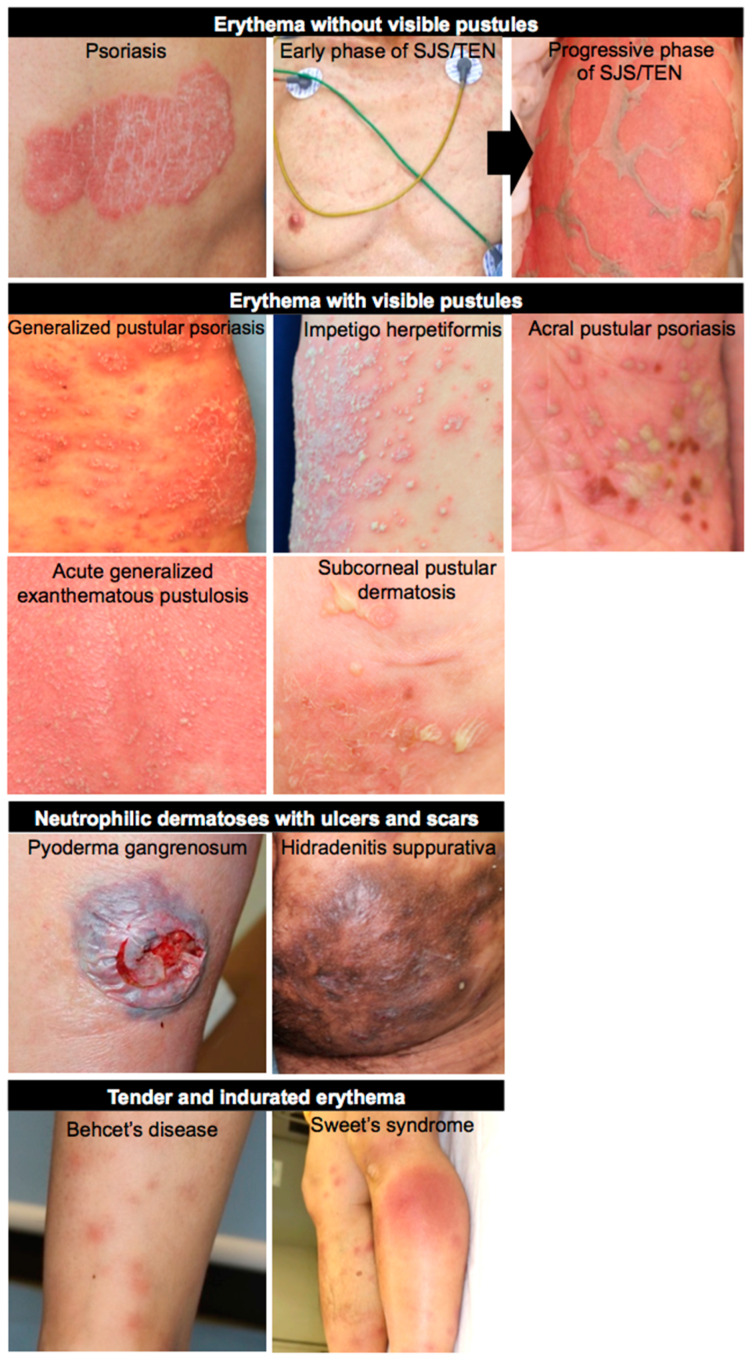
Clinical images of neutrophil-associated skin diseases. Neutrophil-associated dermatological diseases are clinically categorized into four groups: erythema without visible pustules (psoriasis and SJS/TEN); erythema with visible pustules (GPP, IH, APP, AGEP, and SPD); neutrophilic dermatoses with ulcers and scars (PG and HS); tender and indurated erythema (BD and Sweet’s syndrome).

**Figure 2 biomedicines-09-01888-f002:**
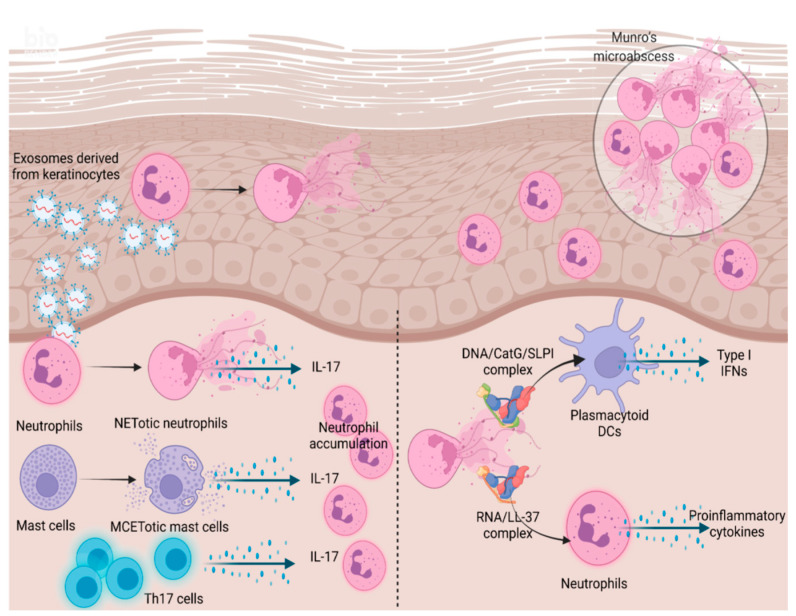
Neutrophil involvement in the pathomechanism of psoriasis. Exosomes derived from human epidermal keratinocytes treated with psoriasis-related cytokines stimulate normal human neutrophils, followed by NET formation. The major IL-17-producing cells in psoriasis might be Th17 cells. However, ETotic neutrophils and mast cells can reportedly serve as a source of IL-17, followed by further neutrophil accumulation. NET-derived DNA and RNA form complexes with CatG/SLPI and LL-37, respectively. The former and latter stimulate pDCs and neutrophils, respectively, followed by production of type I IFNs and proinflammatory cytokines, respectively.

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
