# Peer review of "Neutrophil Extracellular Traps in Skin Diseases"

_biomedicines, 2021, doi:10.3390/biomedicines9121888_

Round 1
Reviewer 1 Report
Ogawa et al. give a concise and demanding review on neutrophil extracellular traps with a focus on skin diseases.
Following comments.
Page 2, line 2: please cite the year of publication (1996)
Figure 1: the order of fotographs is not really evident, the authors define four groups of neutrophilic dermatoses.
Page 4, line 27: with first reading the abbreviation ET was somewhat confusing, extracellular nets is meant, NET poduced by neutrophils, MCET by mastcells, please amply introduce ET,
page 4, line 27: is ETotic neutrophils a terminus technicus?
Page 7, line 17: „there are two possibilities for this phenomenon“ should better read „explantations“.
Page 7, M. Behcet: what is the role of apremilast in netosis? This is the only specifically licenced drug for M. Behcet.
Page 8, line 18ff: there are more data on NETs in Sweet’s syndrome. Please refer to these as well.
Conclusion: how does NETosis differentially result in various clinical pictures? Any qualitative or quantitative explanatory data?
Author Response
Regarding specific comments of Reviewer: 1
- Ogawa et al. give a concise and demanding review on neutrophil extracellular traps with a focus on skin diseases.
Response: We highly appreciate the comments and suggestions made by the Reviewer 1 to improve our manuscript.
- Page 2, line 2: please cite the year of publication (1996)
Response: We have added the publication year as follows: “NETs were first reported in the literature by Takei et al. in 1996[6]”.
- Figure 1: the order of fotographs is not really evident, the authors define four groups of neutrophilic dermatoses.
Response: We appreciate the thoughtful suggestion. We have revised the order of images and defined four groups.
- Page 4, line 27: with first reading the abbreviation ET was somewhat confusing, extracellular nets is meant, NET poduced by neutrophils, MCET by mastcells, please amply introduce ET.
Response: Thank you for pointing this out. We have added the definition and explanation about ET in the section of “2. Neutrophil extracellular traps” as follows: “As described here, NETosis was first termed based on the origin of cells (neutrophils). However, eosinophils, mast cells, monocytes, and macrophages can also release extracellular traps (ETs). Thus, cell death with release of ETs was renamed as ETosis [1,2].”
- Page 4, line 27: is ETotic neutrophils a terminus technicus?
Response: Thank you for pointing this out. When we searched the term “NETotic neutrophils” in Google Scholar, 156 articles were identified. Therefore, we supposed that the term is a terminus technicus, although there is room to discuss.
- Page 7, line 17: „there are two possibilities for this phenomenon“ should better read „explantations“.
Response: Thank you for pointing this out. We have corrected as follows: “There are two possibilities for explanations.”
- Page 7, M. Behcet: what is the role of apremilast in netosis? This is the only specifically licenced drug for M. Behcet.
Response: We appreciate the important comment. There is no published article regarding the role of apremilast in NETosis. Indeed, we had examined the effect of apremilast on NETosis. Apremilast as well as PGE2, a known NET inhibitor, suppressed PMA-induced NET formation in a dose-dependent manner. Unexpectedly, dexamethasone and cyclosporine were unable to suppress NET formation. These data imply that an administration of apremilast suppressesde novoNETosis. We are now under preparation of manuscript. Thus, we have added sentences as follows: “Apremilast, a small molecule inhibitor of phosphodiesterase 4, is the only specifically licensed drug for the treatment of BD. Hence, the effect of apremilast on NET formation should be addressed.”
- Page 8, line 18ff: there are more data on NETs in Sweet’s syndrome. Please refer to these as well.
Response: Thank you for pointing this out. We have added a sentence as follows: “Partial colocalization of IL-1β and TNF-α in NETs of Sweet’s syndrome implies their contribution into NET formation [69].” Additionally, we have found another published article, in which NETs were evaluated in patients with Sweet’s syndrome, and thus cited the article as Reference 70 in the revised manuscript.
- Conclusion: how does NETosis differentially result in various clinical pictures? Any qualitative or quantitative explanatory data?
Response: The question is one of the biggest issues that we have to address in future. As we described in the section of Conclusion, the underlying immunological differences leading to the different clinical features are still unknown. That may be caused by phenotypical difference of infiltrating neutrophils (PMN- or LDG-like phenotype), differential expressions of neutrophil-associated chemokines, or differential processes of NET formation. Unfortunately, we do not have any scientific data to answer the critical question. We appreciate the thoughtful comment.
Reviewer 2 Report
The authors presented a very important topic that has been discussed in recent years. A very important advantage of the manuscript is to organize the knowledge on the subject in a synthetic and understandable way for the reader.
I have read the manuscript with great pleasure
Author Response
Regarding specific comments of Reviewer: 2
The authors presented a very important topic that has been discussed in recent years. A very important advantage of the manuscript is to organize the knowledge on the subject in a synthetic and understandable way for the reader.
I have read the manuscript with great pleasure
Response: We deeply appreciate the positive and kind comments made by Reviewer 2.